# Controlling toxic and harmful gas in blasting with an inhibitor

**Haibao Yi** [1,2,3]*, **Xiliang Zhang**[1,2], **Haitao Yang**[1,2], **Longfu Li**[1,3], **Yu Wang**[1,2], **Sibo Zhan**[1,3]

**1** State Key Laboratory of Safety and Health for Metal Mines, Maanshan, China, **2** Sinosteel Maanshan General Institute of Mining Research Co., Ltd., Maanshan, China, **3** Huawei National Engineering Research Center of High Efficient Cyclic Utilization of Metallic Mineral Resources Co., Ltd., Maanshan, China

* hang_tianfeiji@126.com

## Abstract

In engineering blasting, while efficiently breaking rocks with explosives, a large amount of toxic and harmful gases are generated, which not only pollutes the production environment but also easily leads to explosion smoke poisoning accidents. It must be highly valued by engineering technicians and management personnel. To effectively control the production of harmful gases during explosive blasting, an environmentally friendly and efficient harmful gas inhibitor has been developed, and its mechanism of action has been analyzed and revealed. Through model and on-site experiments, the appropriate addition ratio and charging structure scheme were determined, and good control effects were achieved. The research results indicate that the environment in which explosives are used has a significant impact on the composition of harmful gases produced during blasting. CO, NO, and $NO_2$ are mainly produced in natural air environments, while $NH_3$, CO, and NO are mainly produced in underground blasting environments. As the proportion of inhibitors added increases (2%, 4%, 6%), the decrease in the concentration of harmful gases during blasting first increases and then decreases. Compared with the control experiment, the total reduction rate of harmful gas concentration is 39.23%, 68.20%, and 59.69%, respectively, and the best control effect is achieved when 4% is added. When using the developed inhibitor adding device for the full hole addition scheme, the control effect of harmful gas concentration in blasting is the best, and the decrease in harmful gas concentration reaches 62.79%~84.73% at a distance of 30m~120m. The use of harmful gas inhibitors for blasting combined with other control measures can significantly improve the blasting operation environment, enhance the safety level of production operations, and have good promotion and application value.

## Introduction

Currently, explosive blasting is the most efficient and economical method for crushing ore and rock, and has been widely used in engineering blasting. Due to the influence of explosive composition, explosive reaction degree, rock mass medium, working environment, etc., a large amount of toxic and harmful gases will inevitably be generated during explosive rock breaking, commonly known as blasting smoke, mainly consisting of CO, $NH_3$, and nitrogen oxides

developing research plans, purchasing experimental materials and monitoring instruments, and analyzing data. The authors acknowledge the financial support of this work.

**Competing interests:** NO authors have competing interests.

(NO, $NO_2$, etc.) [1–6]. If sulfur or sulfides are present in explosives or rocks, toxic gases such as hydrogen sulfide and sulfur dioxide will also be generated.

The higher the concentration of toxic gases in the blasting smoke, the greater the gas toxicity and the greater the harm to human health. At the same time, blasting smoke not only pollutes the working environment, but also endangers the occupational health and safety of workers, especially during underground space blasting, which is prone to causing group deaths and injuries. Blasting smoke poisoning accidents must be highly regarded by engineering and management personnel.

In order to effectively control the harmful gas hazards caused by blasting, domestic and foreign scholars have conducted a large amount of related research work from theoretical analysis, model experiments, numerical simulations, and other aspects, and have achieved certain research results. Zawadzka-Małota [7] conducted experimental measurements on the explosive gas composition of various explosives currently produced in Poland. A V Zvyagintseva et al [8]. qualitatively and quantitatively evaluated the harmful gas composition in large-scale blasting of quarries. Rudakov Marat [9] proposed the way of decreasing emission of nitrogen oxides using highly active catalysts as a part of the profiled tamping. And the results obtained showed that zinc carbonate ($ZnCO_3$) is the most effective. Menéndez Javier [10] used a one-dimensional mathematical model and a three-dimensional CFD(Computational Fluid Dynamics) numerical simulation method to analyze the concentration, propagation, and dilution of blasting smoke under different operating conditions. Luo Zhouquan [11] used the computational fluid dynamics simulation software Fluent to study and obtain the time-space trace distribution law of $NO_2$ and CO produced by shaft excavation blasting. YE Yongjun [12] proposed a calculation method for the theoretical minimum ventilation time of radon and blast smoke discharge in single head tunnels by establishing a calculation model. Zhang Xiliang [13] has developed an emulsion explosive explosion gas inhibitor and analyzed the influence of inhibitor addition ratio and loading method on harmful gases. Cao Yang [14] analyzed the diffusion law of CO in the mining face, optimized ventilation parameters, and proposed a control plan for the concentration of blasting smoke. Jain S [15] conducted ammonia monitoring work on the nanocomposites prepared by the research institute (PPy, PPy ZnO, and CSA doped PPy ZnO). Jain S [16] used $NO_2$ sensors for gas monitoring while studying the reactions of nanocomposites with various oxidation and reduction gases. Zhou Q [17] evaluated the gas sensor applications of synthesized Ni and Zn doped $SnO_2$ nanomaterials at different operating temperatures and CO gas concentrations.

In order to reduce the production of toxic and harmful gases during blasting and improve the production and blasting environment, this article develops an efficient blasting harmful gas inhibitor based on the analysis of the mechanism of the inhibitor's action. The appropriate addition ratio and charging structure were determined through experiments, achieving good on-site test results. Compared with previous research, the blasting harmful gas inhibitor developed in this article is a mixture of multiple elemental substances, oxides, steady-state agents, and water, with more complex components. It overcomes the shortcomings and shortcomings of previous inhibitors with single components and less significant harmful gas control effects, and can achieve more obvious harmful gas suppression effects. It provides an effective technical solution for controlling harmful gases in mining blasting. It has a certain guiding and reference role in engineering blasting.

## The mechanism of harmful gas inhibitors in blasting

The developed explosive harmful gas inhibitor is a mixture of various elemental substances (using letters A, B, D instead), oxides ($A_xO_y$, $B_xO_y$, $D_xO_y$), steady-state agents, and water. The

stabilizing agent is an organic and environmentally friendly colloidal material, ensuring uniform mixing of various substances. The stabilizing agent here can promote the uniform dispersion of inhibitor particles in the medium, including types such as water glass, cellulose derivatives, polyacrylamide, and gum. During the experiment, the stabilizing agent, inhibitor, water, and explosives were placed together in a plastic cup, and a glass rod was used to continuously stir until the stirring was uniform, ensuring effective mixing of various substances.

Explosive explosions produce high-temperature and high-pressure gases at the moment. Explosion temperature refers to the heat released during the explosion of an explosive, which heats the explosive product to the highest temperature. Different types of explosives have different explosion temperature ranges, usually between 2000~3000°C. Detonation pressure is the pressure on the detonation front of an explosive during explosion, and its value mainly depends on the chemical properties of the explosive, usually ranging from 10 to 40GPa. Under high temperature and pressure conditions, the inhibitor undergoes chemical reactions with $CO$, $NH_3$, and nitrogen oxides in the blast smoke, generating non-toxic gases such as $CO_2$ and $N_2$, as well as harmless solid substances, greatly reducing the production of toxic and harmful gases during blasting and avoiding the occurrence of blast smoke poisoning accidents.

The principle of action between inhibitors and harmful gases during blasting is shown in Fig 1. For the control of carbon monoxide ($CO$), ammonia ($NH_3$), and nitrogen dioxide ($NO_2$), the inhibitory mechanisms are described as follows.

For carbon monoxide ($CO$), the following oxidation-reduction reactions occur with oxides ($A_xO_y$, $B_xO_y$, $D_xO_y$) to oxidize CO to carbon dioxide ($CO_2$). The generated elemental substances (A, B, D) can continue to provide reducing agents for $NO_2$ control, forming a closed loop. Through this method, CO can be converted into $CO_2$, reducing the production of CO.

$$CO + A_xO_y \left( or\ B_xO_y\ or\ D_xO_y \right) \rightarrow A(or\ B\ or\ D) + CO_2 \tag{1}$$

For ammonia ($NH_3$), the following redox reactions occur with oxides ($A_xO_y$, $B_xO_y$, $D_xO_y$) to oxidize $NH_3$ to $N_2$. Through this method, it is possible to convert $NH_3$ into $N_2$, reduce the

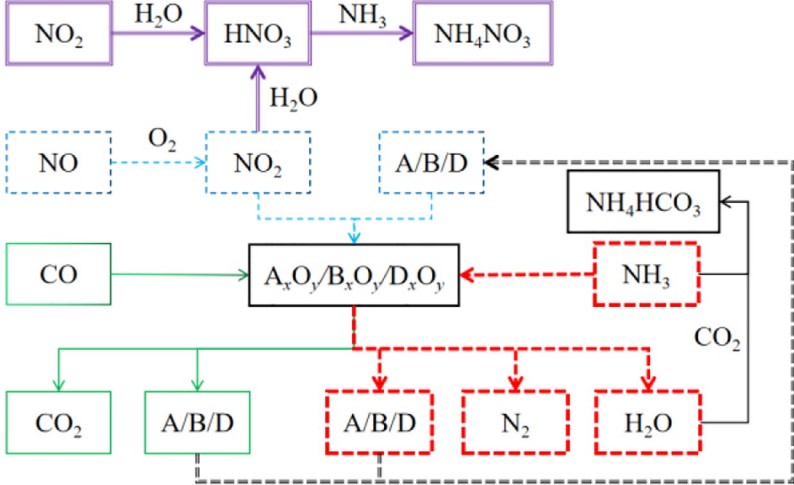

**Fig 1. Schematic diagram of the action principle of inhibitors and harmful gases during blasting.**

production of $NH_3$, and generate $N_2$ and $H_2O$ that are harmless to the air environment.

$$NH_3 + A_xO_y \left( \text{or } B_xO_y \text{ or } D_xO_y \right) \rightarrow N_2 + A(\text{or B or D}) + H_2O \tag{2}$$

For nitrogen dioxide($NO_2$) the following oxidation-reduction reactions occur with elemental substances (A, B, D) to reduce $NO_2$ to $N_2$. The generated oxides ($A_xO_y$, $B_xO_y$, $D_xO_y$) can continue to provide oxidants for CO and $NH_3$ control. In subordinate chemical reactions, $NO_2$ can be reduced to $N_2$, which is harmless to the air environment, effectively reducing the amount of $NO_2$ generated.

$$NO_2 + A(\text{or B or D}) \rightarrow N_2 + A_xO_y \left( \text{or } B_xO_y \text{ or } D_xO_y \right) \tag{3}$$

In addition, the NO and $NH_3$ generated by the explosion also undergo the following chemical reactions, further reducing the concentration of NO and $NH_3$. In subordinate chemical reactions, NO and $NH_3$ can be converted into $NH_4NO_3$ or $NH_4HCO_3$, which effectively reduces the generation of NO and $NH_3$ and plays a role in protecting the environment.

$$NO + O_2 \rightarrow NO_2 \tag{4}$$

$$NO_2 + H_2O \rightarrow HNO_3 \tag{5}$$

$$NH_3 + HNO_3 \rightarrow NH_4NO_3 \tag{6}$$

$$NH_3 + CO_2 + H_2O \rightarrow NH_4HCO_3 \tag{7}$$

Under the action of various chemical reactions mentioned above, the concentration of toxic and harmful gases such as CO, $NH_3$, and nitrogen oxides can be effectively reduced, the harm of blasting harmful gases can be greatly weakened, a good production and operation environment could be created, and the health and safety of production personnel can be maintained.

As shown in Fig 1, the different colors represent different chemical reaction processes, corresponding to the chemical reactions between the explosive toxic gases $NO_2$, NO, CO, $NH_3$ and the inhibitor, achieving the control of these four harmful gases.

Of course, due to the instantaneous and unmeasurable nature of explosive explosion reactions, there may also be additional chemical reactions, such as $NH_3+N_2+H_2 \rightarrow NH_4NO_3$. Due to the lack of effective monitoring instruments for detecting the explosion process, the author believes that there is a certain possibility of this reaction. $NH_4NO_3$ is highly soluble in water, prone to moisture absorption and agglomeration, and can decompose explosively under strong impact or heat. It is mainly used as fertilizer, industrial and military explosives. It has a certain degree of irritation to the eyes, respiratory system, and skin of the human body. Due to the unstable heating characteristics of $NH_4NO_3$, it is believed that $NH_4NO_3$ will not exist in the final product of the explosion and may only exist as a product of the explosion process.

## Model experiment on reducing harmful gases in blasting with inhibitors

To study the effect of blasting harmful gas inhibitors on reducing the concentration of harmful gases, a closed container for blasting experiments was developed, in which explosive explosion tests were conducted. The container is made of 10mm steel plate long × wide × Height = 1m × 1m × 1.5m, and weighs approximately 800kg.

A ventilation door is installed on the front of the container. There is a hole with a diameter of 8mm on the top surface for hanging detonators and explosives. The measurement holes are set on the other three sides with a diameter of 20mm, as shown in Fig 2. In Fig 3, the X-am5000 portable composite gas detector can effectively measure the concentration of harmful substances such as CO, $H_2S$, $CO_2$, $NH_3$, $NO_2$, and $SO_2$. Size: (length × wide × Height) 147mm × 129mm × 31mm, Weight: 220 grams, Environmental conditions: Temperature: -20~+50˚C, Pressure: 700~1300hPa, Humidity: 10%~95% RH. In Fig 4, the emulsion explosive is on the left, a mixture of explosive and inhibitors is on the right. They are formed by stirring and mixing evenly with a glass rod. In Fig 5, the detonator is in the above. A mixture of explosive and inhibitors, wrapped in a white paper is in the below.

## (1) Comparative experiment

In order to understand the composition and concentration of harmful gases generated after the initiation of emulsion explosives, three control experiments were conducted without the addition of explosive harmful gas inhibitors. The amount of explosives used in each experiment was 20g. The test results are shown in Table 1.

From Table 1, it can be seen that without the addition of inhibitors, the average concentration of toxic gases generated after the explosion of 20g emulsion explosives is in the order of CO (105.32ppm)>NO (50.37ppm)>$NO_2$ (32.82ppm). The highest concentration of CO is generated during the explosion of emulsion explosives, followed by NO.

The production process of emulsion explosives adopts a large-scale production process, and various components of the explosives inevitably have certain non-uniformity during the stirring preparation process, which has a certain impact on the performance of the explosives and the production of harmful gases. The different properties of explosives, the varying degrees of response to explosive explosions, and the different usage environments can all have a certain impact on the production of toxic and harmful gases. It was precisely considering these factors that the author conducted three experiments, taking the average of the three

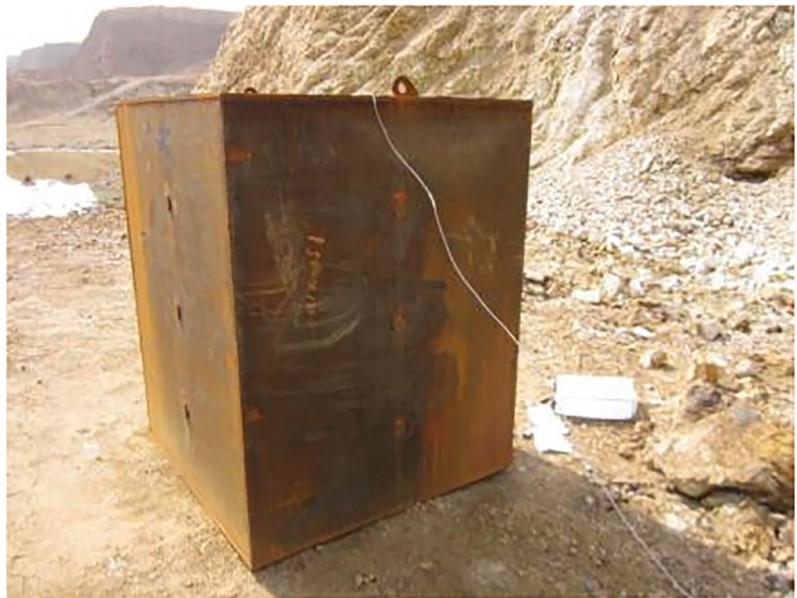

**Fig 2. Closed container diagram.**

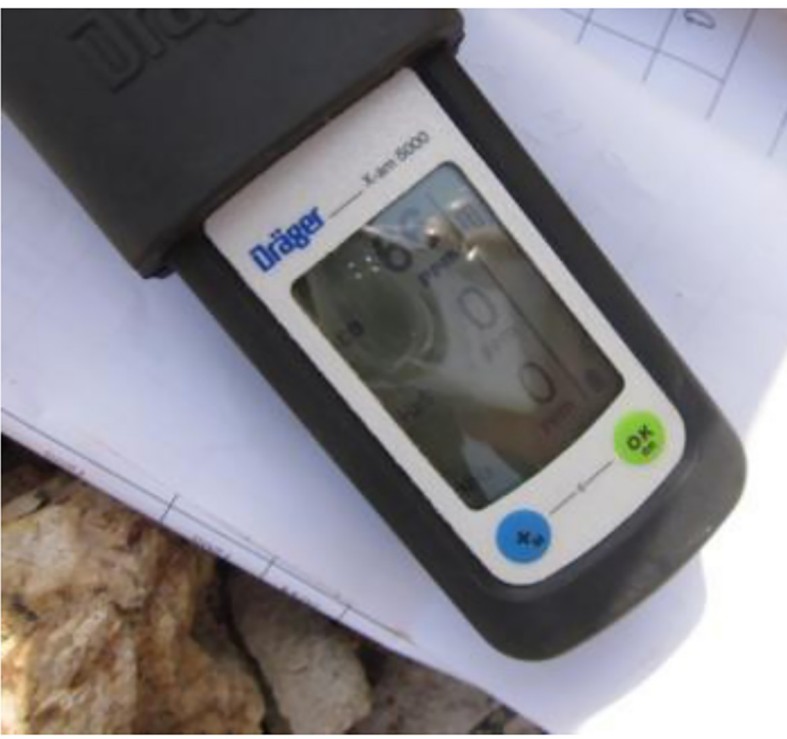

**Fig 3. X-am5000 detector.**

experiments to overcome the impact of uneven explosives and unstable performance on the experimental results. The high concentration of NO in the third test is consistent with the uneven distribution of explosives. After averaging the three test zones, this effect can be eliminated overall.

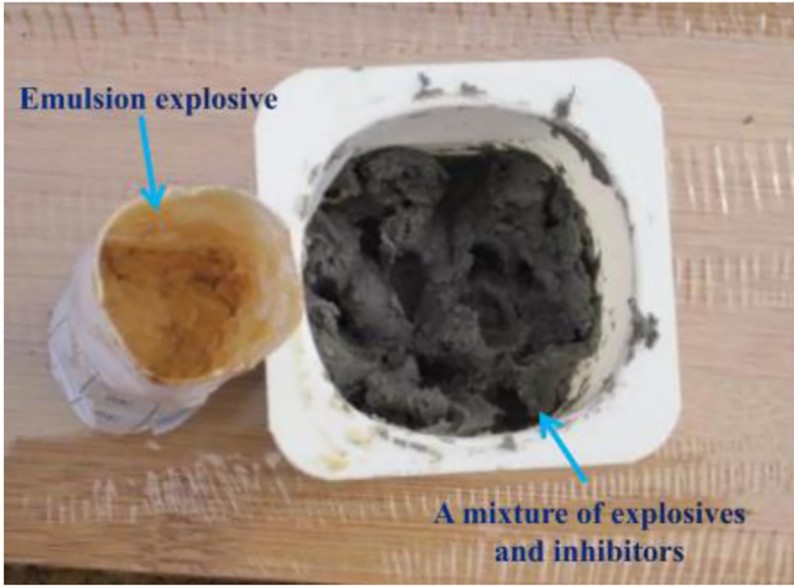

**Fig 4. Experimental materials.**

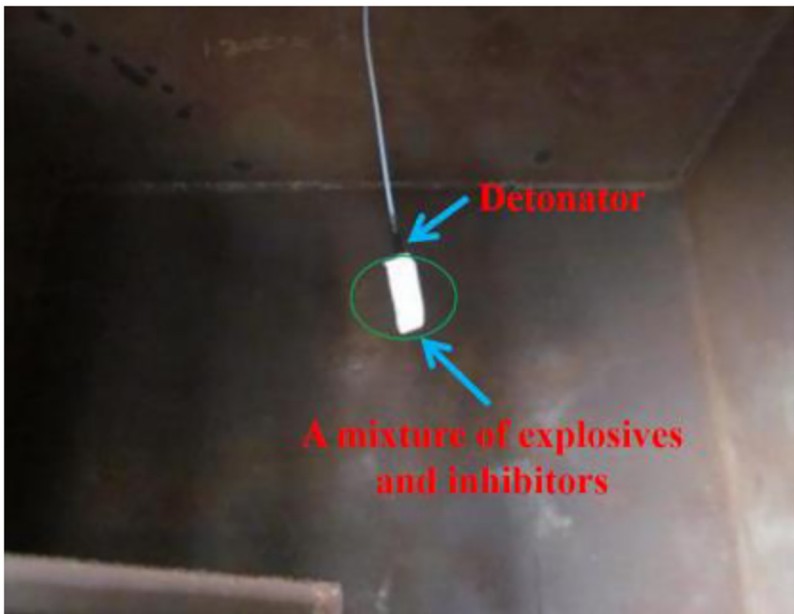

**Fig 5. Experimental photo.**

Based on this calculation, each kilogram of emulsion explosive can produce CO concentration of 5266.17ppm, NO concentration of 2518.33ppm, and $NO_2$ concentration of 1641.00ppm, totaling 9425.50ppm.

Meanwhile, $NH_3$, $SO_2$, and $H_2S$ were not detected during the experiment. Analysis shows that under natural air conditions (with normal temperature 20~25˚C), when there is sufficient oxygen, the explosive explosion reaction is sufficient. $NH_3$ is unstable under high temperature conditions and will decompose into nitrogen and hydrogen, so it does not produce $NH_3$. There is no sulfur element in the explosive composition, so it will not produce $SO_2$ or $H_2S$.

## (2) Inhibitor addition test

Three experiments were carried out on the addition of explosive harmful gas inhibitors based on 2%, 4%, and 6% of the mass of emulsion explosives. The control effects of inhibitor addition ratio on explosive harmful gas were and compared and analyzed. The test results of average value are shown in Table 2, Figs 6 and 7.

From Table 2, Figs 6 and 7, it can be seen that with the increase of the proportion of blasting harmful gas inhibitors added, the concentration of toxic and harmful gases shows a trend of first decreasing and then increasing, and the gas concentration reduction rate shows a pattern of first increasing and then decreasing. The author analyzes and believes that different

**Table 1. Concentration values of harmful gases in explosion of control test.**

| Number | Carbon monoxide (CO)/ppm | Nitric oxide (NO)/ppm | Nitrogen dioxide ($NO_2$)ppm | Total/ppm |
|---|---|---|---|---|
| 1 | 106.62 | 44.35 | 32.27 | 183.24 |
| 2 | 108.01 | 46.43 | 33.07 | 187.51 |
| 3 | 101.34 | 60.32 | 33.12 | 194.78 |
| Average | 105.32 | 50.37 | 32.82 | 188.51 |

**Table 2. Inhibitor addition test data table.**

| Add scale | Carbon monoxide (CO) | | Nitric oxide (NO) | | Nitrogen dioxide (NO₂) | | Total | |
|---|---|---|---|---|---|---|---|---|
| | concentration/ppm | decrease rate% | concentration/ppm | decrease rate% | concentration/ppm | decrease rate% | concentration/ppm | decrease rate% |
| 2% | 89.27 | 15.24 | 18.5 | 63.27 | 6.78 | 79.34 | 114.55 | 39.23 |
| 4% | 55.45 | 47.35 | 4.09 | 91.88 | 0.40 | 98.78 | 59.93 | 68.20 |
| 6% | 68.56 | 34.90 | 5.94 | 88.21 | 1.48 | 95.49 | 75.98 | 59.69 |

inhibitor addition ratios will produce different control effects on harmful gases during blasting, and only when the addition ratio is appropriate can the best control effect be achieved. Visible, the proportion of inhibitors added has a significant impact on the production of harmful gases during explosive explosions, indicating that inhibitors have a significant effect on controlling harmful gases.

When the addition ratio of the inhibitory agent is 2%, 4%, and 6%, the total concentration of harmful gases during blasting is 114.55ppm, 59.93ppm, and 75.98ppm, respectively. Compared with the control experiment, the reduction rate of the total concentration of harmful gases is 39.23%, 68.20%, and 59.69%, respectively. It can be found that when 4% of the inhibitor is added, the concentration of toxic gas is the smallest and the reduction rate is the highest, indicating that the blasting toxic gas control effect is the best at this time. When adding 2%, the concentration of toxic gas is the highest and the control effect is the worst. The effect is centered when adding 6%. Therefore, it is more appropriate to control the inhibitor addition ratio at around 4%.

There is a certain proportional relationship between the amount of inhibitor added and the control effect of harmful gases during blasting, and it is not advisable to add too little or too much. When the amount of inhibitor added is too small, the harmful gas control effect cannot

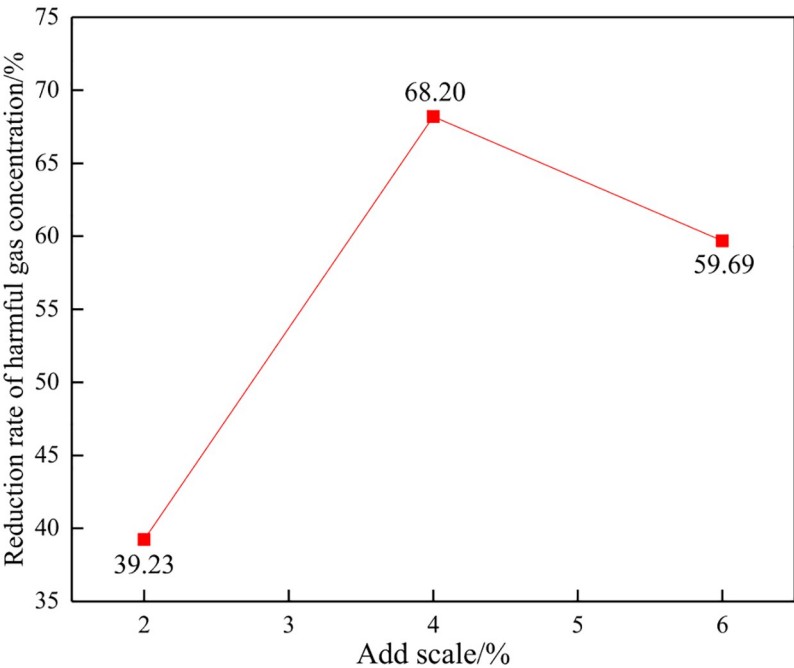

**Fig 6. Reduction in total concentration of harmful gases during blasting.**

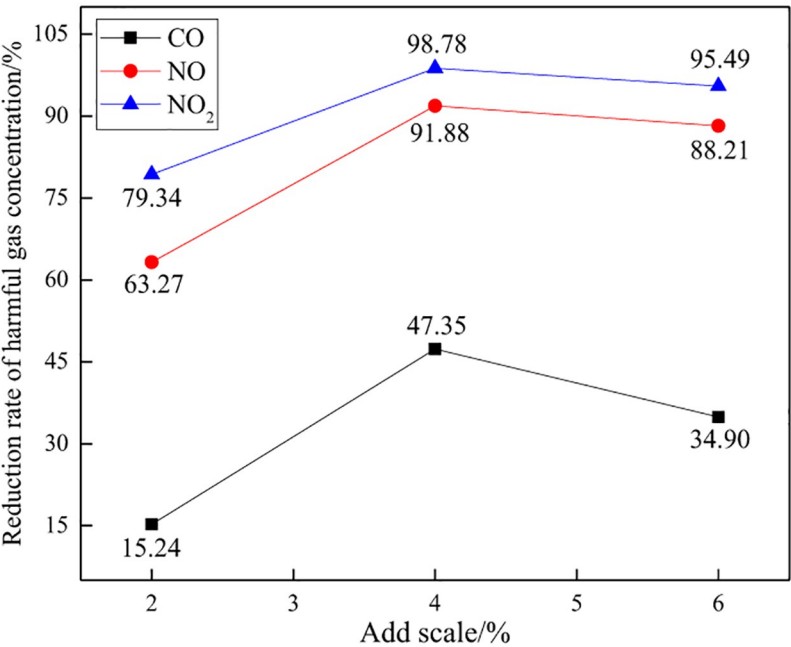

**Fig 7. Reduction rate of various harmful gas concentrations.**

be achieved, and when too much is added, it can cause waste of inhibitor. According to the experiment, it can be found that there is a significant turning point when the inhibitor addition ratio is 4%, at which point the maximum reduction in harmful gases is achieved. Therefore, when the inhibitor addition ratio is greater than 6%, it can be considered that it will cause waste of inhibitors, which is uneconomical.

For various harmful gases, the gas concentration reduction rates of CO, NO, and $NO_2$ are 15.24%~47.35%, 63.27%~91.88%, and 79.34%~98.78%, respectively. It can be seen that the overall control effect of explosive gas inhibitors on NO and NO2 is significantly better than that on CO, indicating that CO control is difficult. In the process of mining production blasting, in order to avoid the occurrence of CO poisoning accidents in mining site blasting, it is necessary to ensure sufficient ventilation and smoke exhaust time, and reduce the CO concentration to the allowable range of on-site personnel operation safety.

## On site testing of harmful gas control during blasting

### Test location

Like other harmful effects of blasting, such as blasting vibration, on-site monitoring is required to determine the concentration of toxic gases during blasting [18–20]. The on-site test site is a sloping excavation working face of an underground iron mine, with a cross-sectional size of 4.4m × 4.0m (width × Height). The on-site engineering geological conditions are complex, and the rock mass structure is relatively developed. The temperature of the test working surface is 38~32°C, the humidity is 80%~86%, and the pressure is one atmospheric pressure.

The CYTJ76 mining hydraulic tunneling drill truck was used for rock drilling, with a diameter of φ = 42mm, hole depth L = 3.3m, 2# rock emulsion explosive, powder roll φ32mm, roll length 300mm, Half second nonel detonator, series parallel detonation network.

## On site blasting test plan

According to the previously determined proportion of harmful gas inhibitors for blasting, on-site blasting tests were conducted on a sloping working face with different addition schemes. Gas concentration collection was carried out using a harmful gas monitor at different distances from the blasting working face. Considering the safety of harmful gas monitoring instruments, in order to avoid damage to the instruments caused by blasting flying rocks and shock waves, it is determined that the distance between the on-site measuring points and the blasting working surface is not less than 30m, and the distance range of the measuring points is 30~100m.

Three inhibitor addition schemes were used in the on-site experiment, including full hole addition, linear addition within the hole, and adding blocked sections of blast holes, as shown in Table 3 and Fig 8. Based on the monitoring data of blasting harmful gases, this study compares and analyzes the control effects of blasting harmful gases under different experiments, and selects the optimal gas inhibitor addition plan to guide the control of blasting harmful gases in mining production operations, improve the quality of blasting operation environment, and maintain the safety of operators.

The explosive harmful gas inhibitor test device developed here is composed of air inlet pipe, air inlet valve, pressure gauge, pneumatic motor, screw pump, feed hopper, discharge valve, flowmeter, discharge pipe, pressure seal joint, blast hole extension rod, etc. The inhibitor is added to the device through the feeding funnel, and is added to the blast hole through this device under the driving force of underground air supply pressure. The outlet diameter of the device is 8mm, the outlet flow rate is V = 1.66m/s, and the outlet flow rate is 83mL/s. The on-site test photos are shown in Figs 9 and 10.

## Monitoring of harmful gases during blasting

In response to the current parameters of slope excavation blasting, on-site monitoring of the concentration changes of toxic and harmful gases during blasting was first carried out [21, 22]. The on-site monitoring data shows that only $NH_3$, CO, and NO were measured in the blasting smoke, while $NO_2$, $SO_2$, and $H_2S$ were all 0, as shown in Fig 11.

The analysis of the paper suggests that the physical properties of CO gas in blasting smoke are relatively stable, and it will not undergo physical and chemical reactions with other substances in the tunnel, and there will be no decrease in mass concentration. However, as the diffusion space becomes larger, the gas concentration will decrease with diffusion dilution.

$NOx$ exists in a mixed gas state dominated by NO and $NO_2$. $NO_2$ is easily soluble in water and reacts with water in underground humid air environments to generate $HNO_3$. There is no sulfur element in explosives and rock masses, so sulfur oxides ($SO_2$, $H_2S$) will not appear.

Based on on-site monitoring data, select a typical curve of toxic and harmful gas concentration change during blasting, and use the function $y = A+B^*x+C^*x^2+D^*x^3$ for regression analysis to obtain the attenuation equation of harmful gas concentration, as shown in Table 4.

**Table 3. Field test scheme table.**

| Number | Inhibitor Addition Scheme | Notes |
|---|---|---|
| Scheme 1 | Full hole Addition Scheme | Add toxic gas inhibitors to the blast hole using a developed experimental device. |
| Scheme 2 | In-hole linear addition scheme | Add a toxic gas inhibitor to the blast hole using a linear sealed plastic tube. |
| Scheme 3 | Scheme for adding blocked sections of blast holes | Use bagged toxic gas inhibitors as gunpowder and place them in the blocked section of the blast hole. |

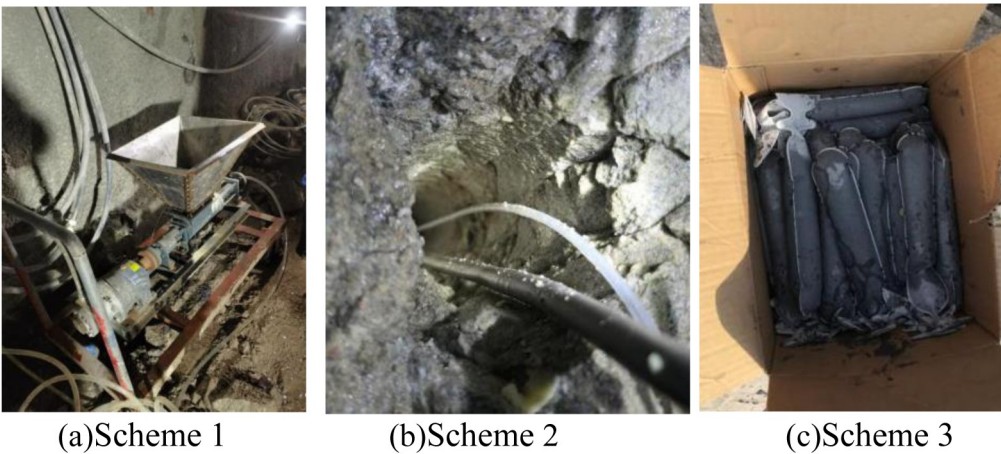

(a)Scheme 1     (b)Scheme 2     (c)Scheme 3

**Fig 8. Field test plan.**

From Fig 11, it can be seen that the peak concentration arrival time of different harmful gases varies slightly, but the overall change pattern is relatively consistent, which can be divided into two distinct stages: the rising stage (steep rising stage) and the falling stage (slow falling stage). Among them, the slope of the rising section is steep, with an approximate straight rise, and the gas concentration quickly reaches its peak. The attenuation in the descending segment is relatively slow and approximates an exponential curve.

As the ventilation time prolongs, the concentration of harmful gases in blasting gradually decreases, and the gas concentration decreases more during the initial stage of ventilation (about 15 minutes), then gradually flattens out. This indicates that the ventilation operation after blasting has a significant effect on diluting the concentration of blast smoke.

Meanwhile, as the distance between the measuring point and the excavation blasting face increases, the peak and total concentration of toxic and harmful gases in blasting show a significant downward trend. Namely, the closer the distance to the blasting working face, the higher

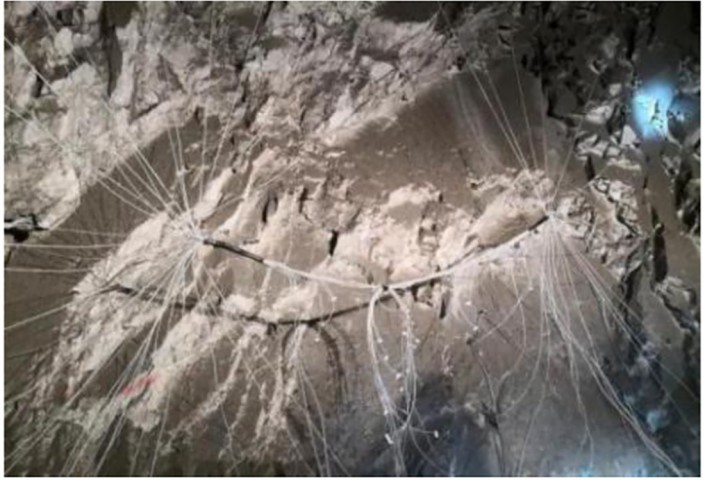

**Fig 9. Connection diagram of blasting network.**

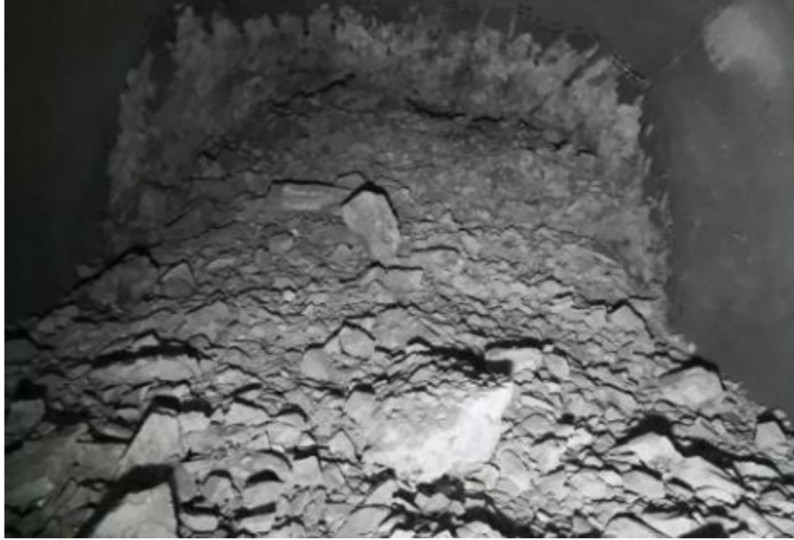

**Fig 10. Field test blasting effect photo.**

the concentration of blasting smoke. The farther the distance, the lower the concentration of blasting smoke. This is consistent with the theory and starting point of blasting smoke diffusion and ventilation smoke exhaust.

Research has shown that the discharge of blasting smoke not only has the transport effect of the main airflow, but also has the turbulent diffusion effect of the airflow, which is a comprehensive process of turbulent deformation and diffusion dilution. As the distance from the blasting working face increases, the volume of blasting smoke distribution gradually expands, and the smoke is constantly replaced with fresh air flow. The fresh air flow carries out high concentration of blasting smoke, gradually reducing the concentration of smoke. Finally, production operations such as support and shovel loading can only be carried out after safety conditions are met.

## Analysis of test results

**(1) Comparison of total gas concentration.** Due to changes in rock mass conditions in mines, the amount of explosives used in each on-site test varies, as shown in Figs 12 and 13.

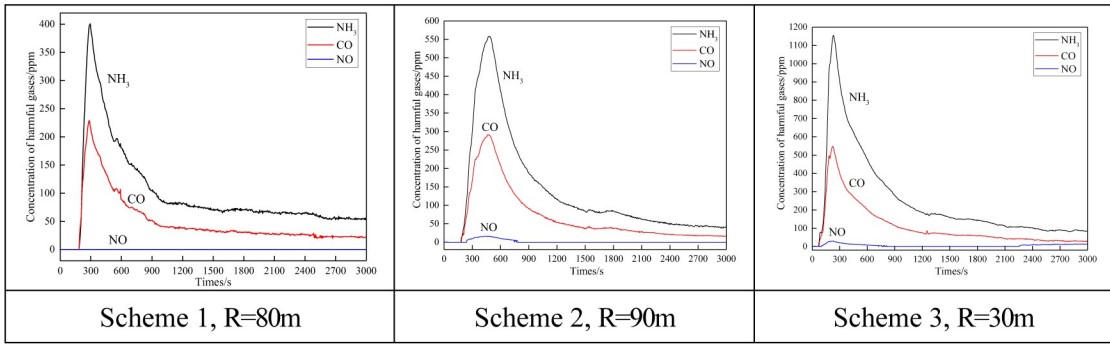

**Fig 11. On site monitoring data of harmful gas concentrations for different schemes.**

**Table 4. Typical curve regression formula.**

| Test number | Gas name | Regression equation | Regression coefficient |
|---|---|---|---|
| Scheme 1, R = 80m | Ammonia (NH$_3$) | $y = 296.303 - 0.358 \cdot x + 1.672E\text{-}4 \cdot x^2 - 2.442E\text{-}8 \cdot x^3$ | $R^2 = 0.921$ |
| | Carbon monoxide (CO) | $y = 165.388 - 0.214 \cdot x + 1.004E\text{-}4 \cdot x^2 - 1.473E\text{-}8 \cdot x^3$ | $R^2 = 0.916$ |
| Scheme 2, R = 90m | Ammonia (NH$_3$) | $y = 464.679 - 0.706 \cdot x + 3.866E\text{-}4 \cdot x^2 - 6.763E\text{-}8 \cdot x^3$ | $R^2 = 0.942$ |
| | Carbon monoxide (CO) | $y = 246.235 - 0.389 \cdot x + 2.149E\text{-}4 \cdot x^2 - 3.790E\text{-}8 \cdot x^3$ | $R^2 = 0.940$ |
| Scheme 3, R = 30m | Ammonia (NH$_3$) | $y = 840.680 - 0.976 \cdot x + 3.982E\text{-}4 \cdot x^2 - 5.120E\text{-}8 \cdot x^3$ | $R^2 = 0.935$ |
| | Carbon monoxide (CO) | $y = 375.019 - 0.446 \cdot x + 1.820E\text{-}4 \cdot x^2 - 2.346E\text{-}8 \cdot x^3$ | $R^2 = 0.946$ |

Therefore, the concentration of toxic and harmful gases in blasting per unit mass of explosives was calculated as the evaluation standard for comparative analysis. At the same time, considering the changes in the distance between monitoring points, the average value of three control experiments is used as a reference to calculate the reduction amplitude of blasting harmful gases for different schemes. The on-site test data is shown in Table 5, Figs 14 and 15.

On site monitoring data shows that, unlike model tests of explosive explosions in natural air environments, there is a significant difference in the composition of toxic and harmful gases produced by explosive explosions during mining slope excavation blasting. The main harmful gases produced at the mining site are NH$_3$ and CO, with concentrations of 17.72ppm/kg and 9.99ppm/kg, respectively, and NO is hardly detected. The environment in which explosives are used has a significant impact on the composition and production of harmful gases.

From the total concentration of harmful gases generated by unit mass explosives, as the distance between the measuring points increases, the concentration of harmful gases in blasting shows an overall decreasing trend. That is, the closer the distance to the blasting working face, the higher the concentration of toxic and harmful gases, while the farther the distance from

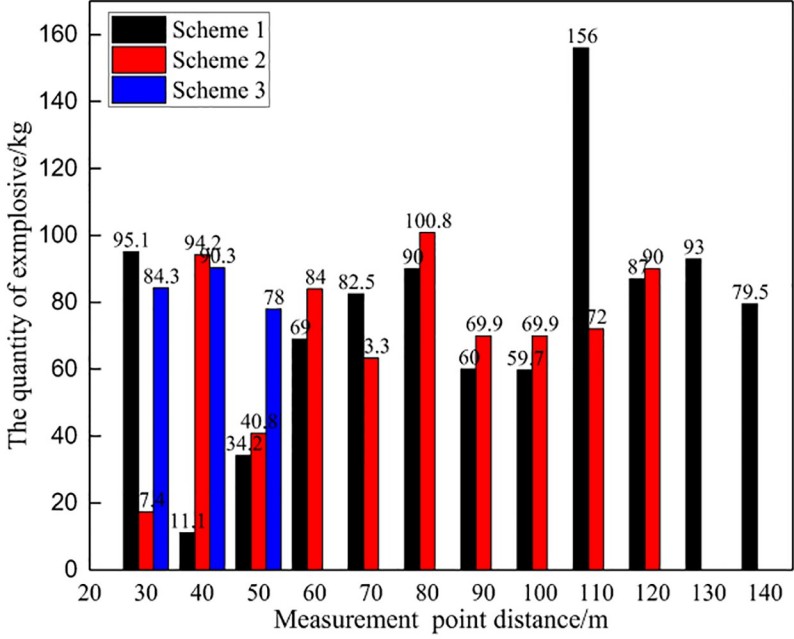

**Fig 12. Explosive usage for different schemes.**

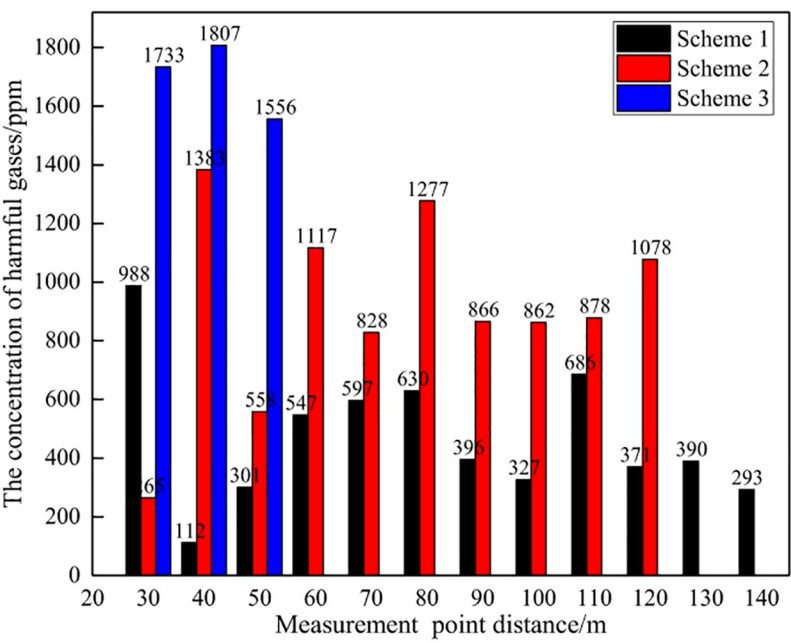

**Fig 13. Toxic and harmful gases generated by different blasting schemes.**

the working face, the lower the concentration of toxic and harmful gases. Analysis suggests that as time goes on, harmful gases from blasting gradually diffuse and dilute, occupying an increasing amount of space, and the gas concentration per unit volume gradually decreases, reflecting a decreasing trend in the concentration of harmful gases.

Meanwhile, at a distance of 30m~120m, the concentration of toxic and harmful gases per unit mass of explosive in Scheme 1 is 4.26~10.39ppm/kg, and the concentration of harmful gases in Scheme 2 is 11.98~15.23ppm/kg. At a distance of 30m~50m, the concentration of harmful gases in Scheme 3 is 19.95~20.56ppm/kg. The experimental results of the three schemes are significantly compared, with Scheme 1 having a significantly lower concentration of harmful gases than the other two schemes, indicating that Scheme 1 can achieve the best harmful gas control effect. It should be noted that Scheme 3 was used to monitor the gas concentration at three different distance points, and the monitoring data was significantly higher than other schemes. It can be determined that this scheme has poor performance, so no further monitoring was conducted at other distances. Therefore, only three measurement point data were used.

From the perspective of the decrease in total concentration of harmful gases, as the distance from the measuring point increases, the decrease in harmful gases during blasting shows an

**Table 5. Control test data table.**

| Number | Distance/m | Peak gas concentration/ppm | | | | Explosive usage/kg | Unit explosive gas concentration(ppm/kg) | | | |
|---|---|---|---|---|---|---|---|---|---|---|
| | | NH$_3$ | CO | NO | Total | | NH$_3$ | CO | NO | Total |
| 1 | 30 | 171 | 87 | 6 | 264 | 9.6 | 17.81 | 9.06 | 0.63 | 27.50 |
| 2 | 30 | 1472 | 873 | 0 | 2345 | 84 | 17.52 | 10.39 | 0.00 | 27.92 |
| 3 | 30 | 1721 | 1016 | 0 | 2737 | 96.6 | 17.82 | 10.52 | 0.00 | 28.33 |
| Average | | | | | | | 17.72 | 9.99 | 0.21 | 27.92 |

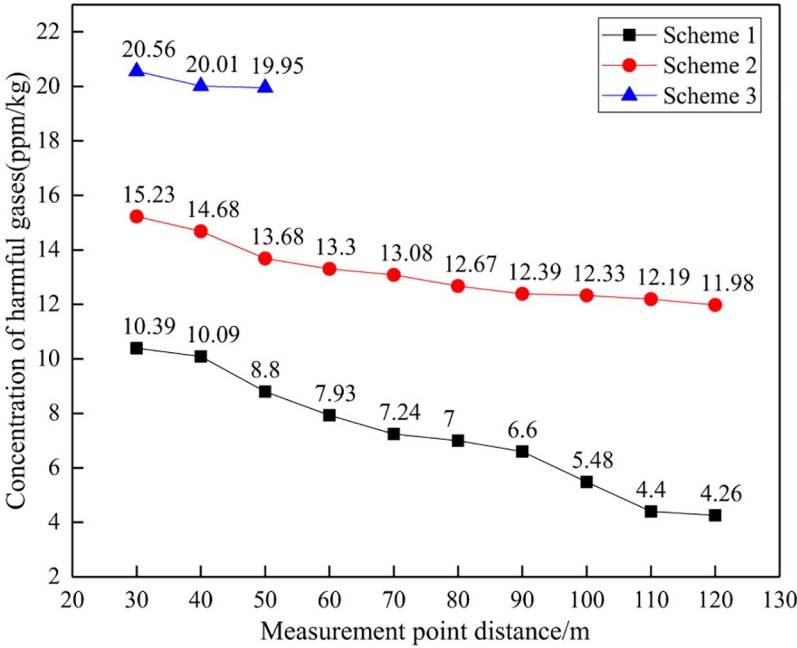

**Fig 14. Concentration of harmful gas in blasting at different distances.**

overall increasing trend. That is, the closer the distance to the blasting working face, the smaller the decrease in harmful gases, while the farther the distance from the working face, the greater the decrease in harmful gases. This is because the comparative analysis is based on the monitoring data at a distance of 30m from the measuring point as the reference standard. The

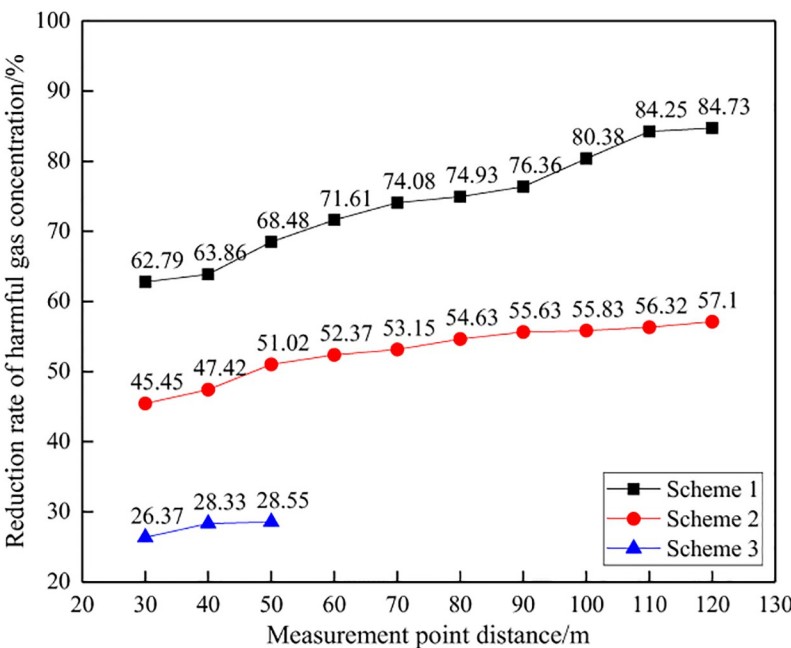

**Fig 15. Reduction rate of blasting gas concentration at different distances.**

farther the distance, the smaller the gas concentration, thus reflecting a greater decrease in magnitude.

Meanwhile, there is a significant difference in the reduction of harmful gases among the three schemes. At a distance of 30m~120m, the decrease in harmful gas concentration in Scheme 1 is 62.79%~84.73%, while in Scheme 2, the decrease in gas concentration is 45.45% ~57.10%. At a distance of 30m~50m, the gas concentration decrease in Scheme 3 is 26.37% ~28.55%. Scheme 1 has the largest reduction in harmful gases, followed by Scheme 2, and Scheme 3 has the worst, indicating that Scheme 1 can achieve the best control effect on harmful gases during blasting.

**(2) $NH_3$ concentration comparison.** The $NH_3$ concentration and reduction effect at different distances are shown in Figs 16 and 17.

From the perspective of the $NH_3$ concentration per unit explosive quantity, as the distance increases, the $NH_3$ concentration shows an overall decreasing trend, which is consistent with the change law of the total concentration of harmful gases. At a distance of 30~120m, the $NH_3$ concentration per unit mass of explosive in Scheme 1 is 2.72~6.90ppm/kg, while Scheme 2 is 7.41~9.89ppm/kg. At a distance of 30m to 50m, the $NH_3$ concentration in Scheme 3 ranges from 12.60 to 13.70ppm/kg. At the same distance, Scheme 1 is the smallest, Scheme 3 is the largest, and Scheme 2 is in the middle. This indicates that Scheme 1 has a greater advantage in controlling $NH_3$ generated by blasting.

From the perspective of the decrease in gas concentration, as the distance between the measuring points increases, the decrease in $NH_3$ concentration shows an overall increasing trend, which is consistent with the change in the total concentration of harmful gases. At a distance of 30~120m, the decrease in harmful gas concentration in Scheme 1 is 61.07%~84.63%, while in Scheme 2, the decrease in gas concentration is 42.22%~58.18%. At a distance of 30~50m, the gas concentration decrease in Scheme 3 is 22.68%~28.88%. There is a significant difference in the decrease in $NH_3$ concentration among different schemes, with the order of decrease

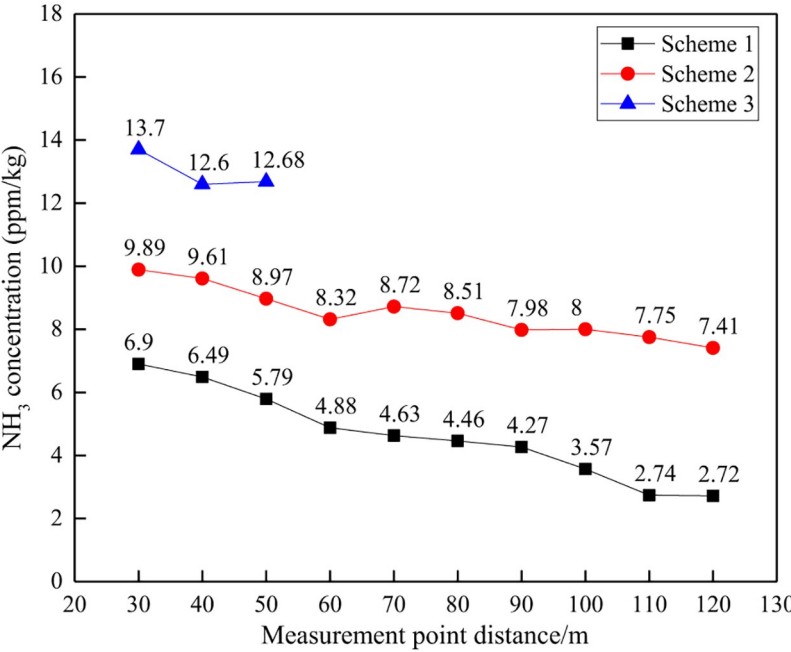

**Fig 16. $NH_3$ concentration at different distances.**

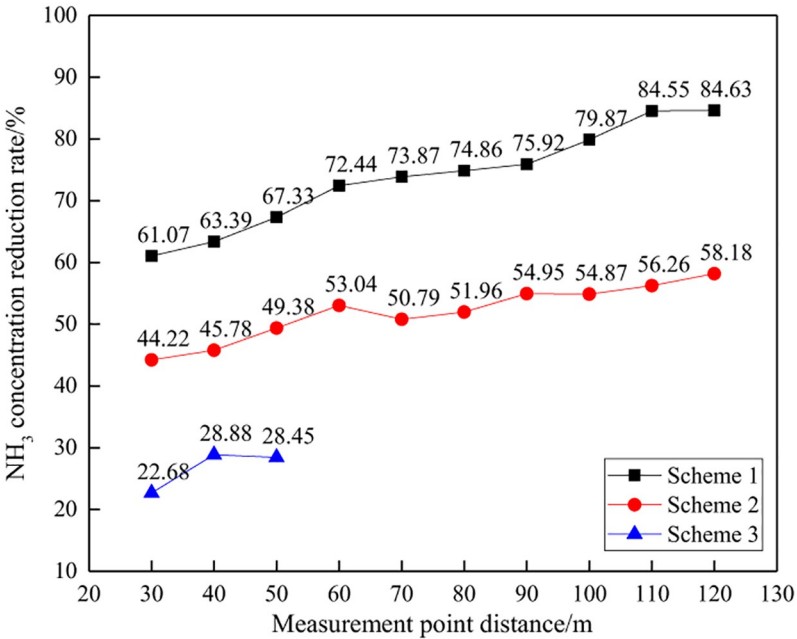

**Fig 17. NH$_3$ concentration decrease at different distances.**

being Scheme 1>Scheme 2>Scheme 3, which also indicates that Scheme 1 has the best NH$_3$ control effect.

**(3) CO concentration comparison.** The CO concentration and reduction effect under different distances are shown in Figs 18 and 19.

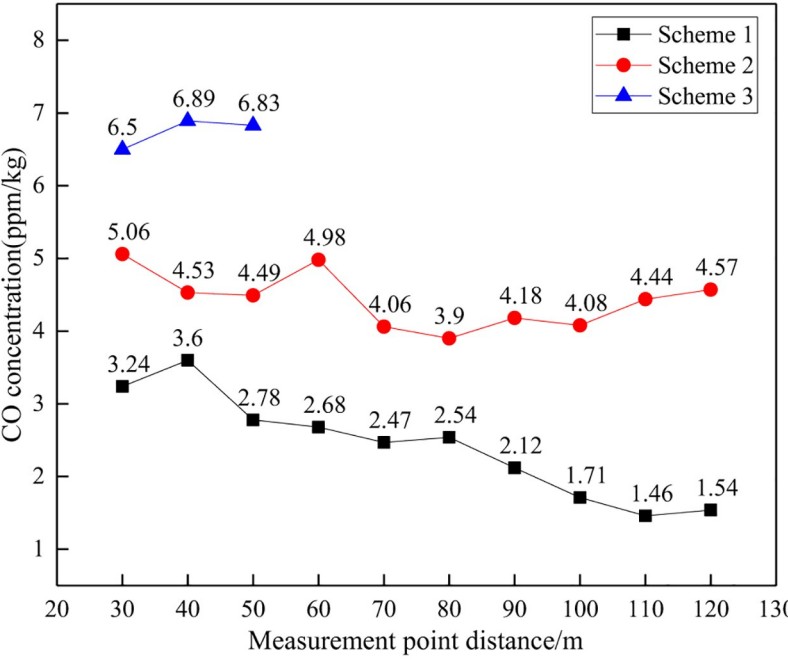

**Fig 18. CO concentration at different distances.**

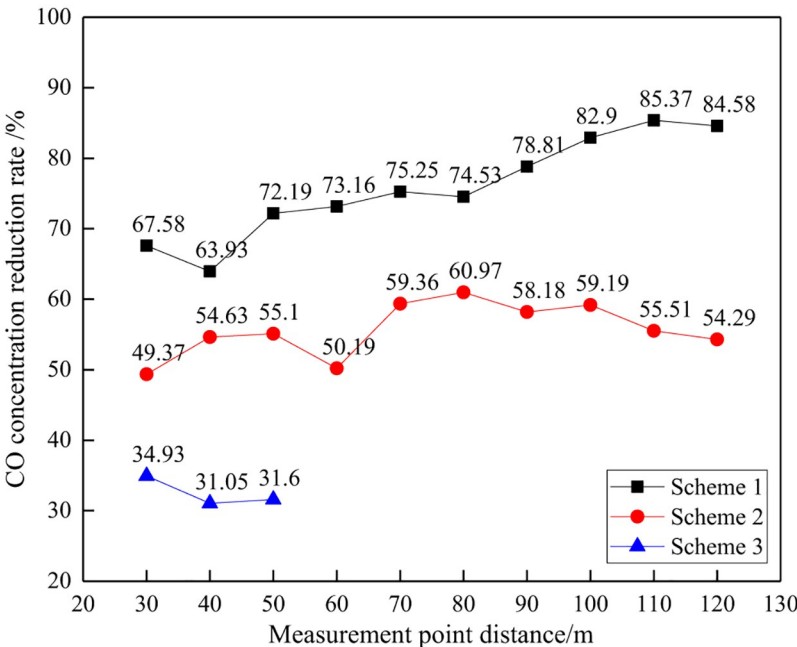

**Fig 19. Reduction rate of CO concentration at different distances.**

From the perspective of CO gas concentration, as the distance between measurement points increases, the overall concentration of CO also shows a continuous decreasing trend. Under the same distance conditions, the order of CO concentration is: Scheme 3>Scheme 2>Scheme 1. From the perspective of the decrease in CO gas concentration, as the distance between measurement points increases, the decrease in CO concentration shows an overall increasing trend. The difference in CO concentration reduction among different schemes reflects the difference in CO concentration control effectiveness. It can be seen that Scheme 1 has the best effect in CO control, followed by Scheme 2.

It should be noted that the variation pattern of CO concentration with distance is generally consistent with that of $NH_3$. There is a certain small fluctuation at local monitoring points, and analysis suggests that it is a normal phenomenon due to the non uniform diffusion process of CO, which is affected by wind and other factors.

## Control measures for harmful gases during blasting

The control measures for toxic and harmful gases in blasting mainly include [23–26]:

1. It is recommended to carry out the promotion and application of toxic and harmful gas inhibitors in blasting, reduce the production of harmful gases from the source of blasting smoke, create a good production and operation environment, and maintain the occupational health and life safety of on-site operators.

2. Select the best type of explosive. The amount of harmful gases produced varies depending on the type of explosive and the blasting environment. According to the type of blasting operation and environmental conditions, select the appropriate explosive type, and try to use zero oxygen balance explosives, which will help reduce the production of harmful gases and improve the environmental quality of the working face.

3. Optimize the parameters of the blasting hole network and strictly control the amount of explosives used for each blasting. There is a positive correlation between the amount of harmful gas produced during blasting and the amount of explosives used. Reducing the amount of explosives used can effectively reduce the amount of harmful gas produced during blasting. Therefore, it is necessary to further optimize the parameters of the blasting hole network and reduce the low explosive dosage while ensuring the quality of the blasting.

4. Using bottom hole initiation technology. When using orifice initiation, the rock at the bottom of the blast hole has a strong clamping effect and the explosive reaction is relatively insufficient, which can easily increase the amount of explosive gas produced. The use of bottom hole initiation can effectively reduce the impact of bottom hole clamping on detonation, improve the completeness of explosive detonation reaction, and reduce the amount of explosive gas produced.

5. Ensure the quality of borehole blockage. The higher the quality of hole blockage, the more it can ensure the degree of explosive detonation reaction. Suitable materials for hole blockage should be selected, water mud blockage method should be used, and skill training for on-site operators should be strengthened to improve the quality of hole blockage, thereby effectively reducing the production of harmful gases.

6. Establish automatic spray system for stope blasting. Some harmful gases from blasting are easily soluble in water. By establishing automatic spray facilities, the concentration of harmful gases can be further reduced during the diffusion of blasting smoke. If necessary, alkaline liquid can be sprayed to reduce the concentration of nitrogen compounds. At the same time, the automatic spray system can also effectively reduce the blasting dust and improve the air quality of the working face.

7. When underground blasting, the type of ventilation fan should be selected to strengthen on-site ventilation. For residual explosive harmful gases that cannot be eliminated, ventilation of the working face shall be adopted to exhaust them. Taking into account factors such as ventilation and smoke exhaust distance, working face size, etc., a suitable type of ventilation fan is selected, and the maintenance of ventilation fans, air ducts, and other equipment is strengthened. After blasting, toxic and harmful gases in the mining area are promptly discharged, creating conditions for subsequent shovel loading, support, and other operations.

## Results and discussion

1. Different explosive usage environments can produce harmful explosive gases with different compositions and concentrations. Blasting harmful gas inhibitors are effective methods for controlling the concentration of blasting harmful gases, which have important practical significance in reducing the production of blasting harmful gases and improving the production and operation environment. It is recommended to promote their application.

2. Different proportions of explosive harmful gas inhibitors can have varying degrees of harmful gas reduction effects. With the increase of inhibitor addition ratio (2%, 4%, 6%), the decrease in the concentration of harmful gases during blasting shows a trend of first increasing and then decreasing. The best control effect of harmful gases during blasting is achieved when the addition ratio is 4%, with a total concentration reduction rate of 68.20%. Compared with references [3, 4, 14], the present research shows that the reduction effect of

harmful gas concentration in blasting is more significant, which can effectively demonstrate the good effect of the blasting inhibitors described in the paper.

3. There is a positive correlation between the amount of charge in the blast hole and the production of toxic and harmful gases during blasting. The larger the amount of explosives used, the greater the production of harmful gases during blasting. Under the conditions of rock fragmentation by blasting, it is advisable to reduce the amount of explosives used to effectively reduce the concentration of harmful gases during blasting.

4. The addition method of blasting inhibitors will also have a significant impact on the reduction of harmful gases during blasting. When using the developed experimental device for inhibitor addition (Scheme 1), the optimal effect of reducing harmful gas concentration in blasting can be achieved. At a distance of 30m~120m, the concentration of harmful gas decreases by 62.79%~84.73%, which has strong guidance for improving the on-site blasting operation environment. Meanwhile, in Scheme 1, a mixture of inhibitor and water is added. Due to the incompressible nature of water, it has a certain hydraulic pressure boosting effect during the blasting process, which can to some extent increase the expansion range of rock fractures, increase the average rock crack rate, and help improve the blasting rock breaking effect.

5. Effective measures to control harmful gases during blasting, such as optimizing explosive types and hole network parameters, using hole bottom initiation technology, and strengthening on-site ventilation, can further reduce the production of harmful gases and improve the quality of the blasting environment. It is suggested that mines should adopt various methods for controlling harmful gases in blasting based on their own conditions to ensure production safety and avoid accidents caused by blasting smoke poisoning.

## Acknowledgments

This work was supported by the National Key R&D Program of China (No. 2022YFC2904101). The funder has played a role in developing research plans, purchasing experimental materials and monitoring instruments, and analyzing data. The authors acknowledge the financial support of this work.

## Author Contributions

**Data curation:** Yu Wang, Sibo Zhan.

**Investigation:** Haitao Yang, Longfu Li.

**Methodology:** Xiliang Zhang.

**Supervision:** Longfu Li.

**Writing – original draft:** Haibao Yi, Xiliang Zhang.

**Writing – review & editing:** Haibao Yi, Haitao Yang.

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
