## [Decision Letter · Decision Letter 0]

28 Jul 2023

PONE-D-23-20758Controlling toxic and harmful gas in blasting with an inhibitorPLOS ONE

Dear Dr. Yi,

Thank you for submitting your manuscript to PLOS ONE. After careful consideration, we feel that it has merit but does not fully meet PLOS ONE’s publication criteria as it currently stands. Therefore, we invite you to submit a revised version of the manuscript that addresses the points raised during the review process.

We look forward to receiving your revised manuscript.

Kind regards,

Niravkumar Joshi

Academic Editor

PLOS ONE

Journal Requirements:

   "This work was supported by the National Key R&D Program of China (No. 2022YFC2904101). The authors acknowledge the financial support of this work."

   "the National Key R&D Program of China (No. 2022YFC2904101)."

   "the National Key R&D Program of China (No. 2022YFC2904101)."

5.Please ensure that you refer to Figures 1, 3, 4, 5, 9 and 10 in your text as, if accepted, production will need this reference to link the reader to the figure.

Reviewers' comments:

Reviewer's Responses to Questions

**Comments to the Author**

1. Is the manuscript technically sound, and do the data support the conclusions?

Reviewer #1: Yes

Reviewer #2: Yes

Reviewer #3: Partly

Reviewer #4: Partly

2. Has the statistical analysis been performed appropriately and rigorously? 

Reviewer #1: Yes

Reviewer #2: Yes

Reviewer #3: Yes

Reviewer #4: I Don't Know

3. Have the authors made all data underlying the findings in their manuscript fully available?

Reviewer #1: No

Reviewer #2: Yes

Reviewer #3: Yes

Reviewer #4: No

4. Is the manuscript presented in an intelligible fashion and written in standard English?

Reviewer #1: Yes

Reviewer #2: Yes

Reviewer #3: Yes

Reviewer #4: No

5. Review Comments to the Author

Reviewer #1: Zhang Xiliang et. al. reported the Controlling toxic and harmful gas in blasting with an inhibitor. The authors have developed an efficient blasting harmful gas inhibitor based on the analysis of the mechanism of the inhibitor's action. The appropriate addition ratio and charging structure were determined through experiments, achieving good on-site test results and providing an effective technical solution for controlling harmful gases in mining blasting. However, a few more clarifications are needed before the manuscript can be considered suitable for publication. A list of other comments that need to be addressed is as follows:

1. Author should discuss more about role of Inhibitor.

2. Amount of charge in a hole and average crack rate should be explained in result and discussion part.

3. Comparison of present study to literature should be included.

4. In the introduction section, the recent development of sensors towards toxic gases like NO, NO2, which can help to provide a full background to the readers, to provoke the readers’ new ideas, and to improv the impact of the present job, some key publications such as Materials science and engineering B, 247, 2019,114381, Ceramic International,44, 2018, 4392, Applied Surface Science, 396 (2017) 1317.

5. Some errors and inappropriate places should be corrected, e.g. such as ---etc; NH3 (line 72) should be NH3.

6. In table 2 and table 4, the gases should be in English language.

Reviewer #2: The author has conducted intensive study to show significance of their work. Minor revisions are required -

1) Line 141, author states "From Table 2, Figure 6, and Figure 7, it can be seen that with the increase of the proportion

143 of blasting harmful gas inhibitors added, the concentration of toxic and harmful gases shows a

144 trend of first decreasing and then increasing, and the gas concentration reduction rate shows a

145 pattern of first increasing and then decreasing." Why is this trend followed is not clearly mentioned by the author.

2) In section 4, author should show the superiority of their work by referencing or mentioning about the effect of other factors such as humidity, temperature, and pressure in their proposed solution.

3) Add references for the statement in Line 230.

4) In section 3, line 307, author should mention why their data for CO concentration vs distance is deviating from the trend as observed in NH3 and other gases concentration comparison.

Reviewer #3: The author of the paper successfully showed the importance of the work and its impact on the environment. However, there are multiple places where some clarification/corrections are required.

1. Author didnt provide details of harmful gas inhibitors (A, B, D). What are these elements ? Is the inhibitor reaction with harmful gases thermodynamically feasible ? Enthalpy of reaction discussion missing ?

2. Line 51 - Full form of CFD ?

3. Line 102 - Short description of Fig 1 in the text to explain the action principal of inhibitors and harmful gas . Is there any significance of grouping the gases in specific colors ?

4. Line 114 - More information required in the text with details

Fig3 - Details fo the X-am detector

Fig4 - What chemicals are these ? How they were prepared ?

Fig5 - Is it detonator and explosives ? If yes then the details are required. The image is not very clear. Recommedn to add magnified image of the white thing with explaination what is it ?

5. Line 120 - Which technique was used to collect the concentration ?

6. Line 130 - What temperature value ? Reference required if lab data is not available

7. Line 138 - Please explain the character used as table heading ?

8. Line 269 - Why Scheme 3 has just 3 points ? How ppm/kg numbers are calculated for different schemes ?

9. Line 315 - Why Scheme 1 slope is different compared to scheme 2 ? What does the slope of the curve signifies and what factors will influence it.

Reviewer #4: To the Authors,

Congratulations on your study aiming to understand the amount of toxic and harmful gases generated during engineering blasting and developing inhibitors to reduce pollutants and smoke poisoning accidents. Your research addresses critical environmental and safety concerns and has the potential to make a significant impact in the field. However, before recommending your paper for publication, there are several points that need clarification and improvement to enhance its scientific rigor and readability:

1. Introduction: While your review of previous works is comprehensive, it lacks clarity on the novelty and differentiation of your study. Please explicitly state how your work differs from previous research, particularly regarding the use of harmful gas inhibition. (Line 30-66)

2. Composition of A, B, D elements: Please provide a clear explanation of the composition for elements A, B, and D, as mentioned in the paper. This information is essential for readers to understand the materials used. (Line 69)

3. Stabilizing Agent: Clarify the specific organic and environmentally friendly stabilizing agent used and elaborate on how it ensures efficient mixing. (Line 70)

4. Reaction Conditions: Include the temperature and pressure conditions at which the inhibitor undergoes the reaction, as this information is crucial for understanding the kinetics of the process. (Line 71)

5. Mechanism of Harmful Gas Inhibitors: In the section explaining the mechanism of harmful gas inhibitors in blasting, address whether additional reactions, such as NH3+N2+H2 → NH4NO3 (or any other byproducts), might be generated. If so, provide information on the toxicity of these byproducts. (Line 67)

6. Dimensions of Closed Container: Clarify the dimensions of the closed container used for the blasting experiment in line 105.

7. Control Experiments: Explain whether all control experiments are identical. If so, please address the discrepancy in Table 1, where the ppm for NO generated in the 3rd test is approximately 33% higher than the average of test 1 and 2. Clarify whether this deviation is expected or requires further investigation. (Line 117)

8. Table 2: Provide clarification for the Chinese character in Table 2.

9. Saturation at 6%: In lines 153-155, explain why the performance saturated at 6%

10. Inhibition Effect Between 4% and 6%: Based on the data, 4% showed the highest reduction in harmful gas concentration. Have you explored the inhibition effect between 4% and 6%? If so, please include those findings.

11. Onsite Testing: For the different schemes of onsite testing, specify the amount of inhibitor used. Was it 4% of the emulsion explosive?

12. Quantity of Emulsion Explosive Used: In line 169, share the quantity of emulsion explosive used for the onsite testing.

13. Scheme 1 Performance: Explain why Scheme 1 performed better than other schemes in the onsite testing. Provide a more comprehensive conclusion to elucidate the mechanism behind why 4% and Scheme 1 yielded superior results.

Addressing these concerns will significantly improve the clarity, accuracy, and impact of your research. Once these revisions are made, I believe your paper will be well-suited for publication. Your work holds substantial potential to enhance overall environmental safety and protect the welfare of workers in the engineering blasting industry.

Best wishes for the successful completion of your revisions and subsequent publication.

6. PLOS authors have the option to publish the peer review history of their article (what does this mean?). If published, this will include your full peer review and any attached files.

Reviewer #1: No

Reviewer #2: **Yes: **Nayna Khosla

Reviewer #3: **Yes: **Nayan Chakravarty

Reviewer #4: No

---

## [Author Response · Author response to Decision Letter 0]

8 Aug 2023

As shown in the file: Response to Reviewers.docx.

---

## [Decision Letter · Decision Letter 1]

4 Sep 2023

Controlling toxic and harmful gas in blasting with an inhibitor

PONE-D-23-20758R1

Dear Dr. Yi Haibao,

We’re pleased to inform you that your manuscript has been judged scientifically suitable for publication and will be formally accepted for publication once it meets all outstanding technical requirements. Please note that comments from Reviewer 4 are not required, but if authors may include any clarifications, please revise and submit the manuscript again.

Kind regards,

Niravkumar Joshi

Academic Editor

PLOS ONE

Additional Editor Comments (optional):

Reviewers' comments:

Reviewer's Responses to Questions

**Comments to the Author**

1. If the authors have adequately addressed your comments raised in a previous round of review and you feel that this manuscript is now acceptable for publication, you may indicate that here to bypass the “Comments to the Author” section, enter your conflict of interest statement in the “Confidential to Editor” section, and submit your "Accept" recommendation.

Reviewer #1: All comments have been addressed

Reviewer #2: All comments have been addressed

Reviewer #3: All comments have been addressed

Reviewer #4: (No Response)

2. Is the manuscript technically sound, and do the data support the conclusions?

Reviewer #1: Yes

Reviewer #2: Yes

Reviewer #3: Yes

Reviewer #4: Partly

3. Has the statistical analysis been performed appropriately and rigorously? 

Reviewer #1: Yes

Reviewer #2: Yes

Reviewer #3: Yes

Reviewer #4: No

4. Have the authors made all data underlying the findings in their manuscript fully available?

Reviewer #1: Yes

Reviewer #2: Yes

Reviewer #3: Yes

Reviewer #4: No

5. Is the manuscript presented in an intelligible fashion and written in standard English?

Reviewer #1: Yes

Reviewer #2: Yes

Reviewer #3: Yes

Reviewer #4: No

6. Review Comments to the Author

Reviewer #1: Conclusion on research article:

Zhang Xiliang et. al. reported the Controlling toxic and harmful gas in blasting with an inhibitor. All the comments are addressed properly, that manuscript is ready for publication.

Reviewer #2: Author addressed the required questions. The manuscript is written in good format and is scientific. It is good to publish.

Reviewer #3: The Author has answered all of the questions. No further clarification required from my side. Thanks

Reviewer #4: To the Authors,

1. The Introduction is still missing the novelty and differentiation of your study (particularly regarding the use of harmful gas inhibition). (Line 30-66)

2. Clarity on the stabilizing Agent and how is ensures efficient mixing is unclear. (Line 70)

3. Dimensions of Closed Container is missing in line 105.

4. Chinese character still exists in Table 2.

5. The updated version doesn’t explain why the performance saturated at 6% (lines 153-155)

6. Quantity of Emulsion Explosive Used: In line 169, the quantity of emulsion explosive used for the onsite testing is missing.

7. Results and conclusion doesn’t provide a comprehensive conclusion to explain why 4% was inhibitor and scheme 1 yielded superior results.

7. PLOS authors have the option to publish the peer review history of their article (what does this mean?). If published, this will include your full peer review and any attached files.

Reviewer #1: No

Reviewer #2: No

Reviewer #3: **Yes: **Nayan Chakravarty

Reviewer #4: No

---

## [Editor Report · Acceptance letter]

26 Sep 2023

PONE-D-23-20758R1 

Controlling toxic and harmful gas in blasting with an inhibitor 

Dear Dr. Yi:

I'm pleased to inform you that your manuscript has been deemed suitable for publication in PLOS ONE. Congratulations! Your manuscript is now with our production department. 

Kind regards, 

on behalf of

Dr. Niravkumar Joshi 

Academic Editor

PLOS ONE